# Skip-gram - Zipf + Uniform = Vector Additivity

## Abstract

In recent years word-embedding models have gained great popularity due to their remarkable performance on several tasks, including word analogy questions and caption generation. An unexpected "side-effect" of such models is that their vectors often exhibit compositionality, i.e., *adding* two word-vectors results in a vector that is only a small angle away from the vector of a word representing the semantic composite of the original words, e.g., "man" + "royal" = "king".

In this work we provide a mathematical formalism for compositionality, and a theoretical justification for the presence of additive compositionality in word vectors learned using the Skip-Gram model. In particular, we show that additive compositionality holds in an even stricter sense (small distance rather than small angle) under certain assumptions on the process generating the corpus. As a corollary, we explain the success of vector calculus for solving word analogies. When these assumptions do not hold, we show that compositionality is no longer additive, and provide the correct composition operator.

Finally, we establish a connection between the Skip-Gram model and the Sufficient Dimensionality Reduction (SDR) framework of Globerson and Tishby. SDR models provide information-theoretically optimal symbol-embeddings for problems where the training data takes the form of co-occurrence statistics. We prove that the parameters of Skip-Gram models can be readily modified to obtain the parameters of SDR models by simply adding information on symbol frequencies. This shows that the Skip-Gram model is essentially learning optimal word embeddings in the sense of Globerson and Tishby. Further, it implies that the heuristics commonly used to approximately fit Skip-Gram models can be used to fit SDR models.

## 1 Introduction

The idea of representing words as vectors has a long history in computational linguistics and machine learning. The general idea is to find a map from words to vectors such that word-similarity and vector-similarity are in correspondence. Whilst vector-similarity can be readily quantified in terms of distances and angles, quantifying word-similarity is a more ambiguous task. A key insight in that regard is to posit that the meaning of a word is captured by "the company it keeps" (Firth, 1957) and, therefore, that two words that keep company with similar words are likely to be similar themselves. To break the cyclicality of this definition it is common to consider models that only attempt to capture pairwise co-occurrence statistics[1].

In the simplest case, one seeks vectors whose similarity approximates the co-occurrence frequencies. In more sophisticated methods co-occurrences are reweighed to suppress the effect of more frequent words (Rohde et al., 2006) and/or to emphasize pairs of words whose co-occurrence frequency maximally deviates from the independence assumption (Church and Hanks, 1990).

An alternative to seeking word-embeddings that reflect co-occurrence statistics is to extract the

---

[1]Normally, if a is similar to b and b is similar to c, it should be that a is more similar to c than it is to a random word. But pairwise-similarity models do not explicitly require that, i.e., do not penalize model parameters under which a and c are not particularly similar.

vectorial representation of words from non-linear statistical language models, specifically neural networks. (Bengio et al., 2003) already proposed (i) associating with each vocabulary word a *feature vector*, (ii) expressing the *probability function* of word sequences in terms of the feature vectors of the words in the sequence, and (iii) learning *simultaneously* the vectors and the parameters of the probability function. This approach came into prominence recently through works of Mikolov et al. (see below) whose main departure from (Bengio et al., 2003) was to follow the suggestion of (Mnih and Hinton, 2007) and trade-away the expressive capacity of general neural-network models for the scalability to very large corpora afforded by the more restricted class of log-linear models.

An unexpected side effect of deriving word-embeddings via neural networks is that the word-vectors produced appear to enjoy (approximate) *additive compositionality*: adding two word-vectors often results in a vector whose nearest word-vector is of the word capturing the composition of the added words, e.g., "man" + "royal" = "king" (Mikolov et al., 2013c). This unexpected property allows one to use these vectors to answer word-analogy questions *algebraically*, e.g., answering the question "Man is to king as woman is to ?" by returning the word whose word-vector is nearest to the vector

$$\mathbf{v}(\text{king}) - \mathbf{v}(\text{man}) + \mathbf{v}(\text{woman}).$$

In this work we focus on explaining the source of this phenomenon for the most prominent such model, namely the Skip-Gram (SG) model introduced in (Mikolov et al., 2013a). The SG model learns vector representations of words based on their patterns of co-occurrence in the training corpus as follows: it assigns to each word $c$ in the vocabulary $V$, a "context" and a "target" vector, respectively $\mathbf{u}_c$ and $\mathbf{v}_c$, which are to be used in order to predict the words that appear around each occurrence of $c$ within a window of $\Delta$ tokens. Specifically, the log probability of any target word $w$ to occur at any position within distance $\Delta$ of a context word $c$ is taken to be proportional to the inner product between $u_c$ and $v_w$, i.e., letting $n = |V|$,

$$p(w|c) = \frac{e^{\mathbf{u}_c^T \mathbf{v}_w}}{\sum_{i=1}^n e^{\mathbf{u}_c^T \mathbf{v}_i}} \ . \qquad (1)$$

Further, SG assumes that the conditional proba-

bility of each possible set of words in a window around a context word $c$ factorizes as the product of the respective conditional probabilities:

$$p(w_{-\Delta}, \dots, w_\Delta | c) = \prod_{\substack{\delta = -\Delta \\ \delta \neq 0}}^{\Delta} p(w_\delta | c). \qquad (2)$$

(Mikolov et al., 2013a) proposed learning the SG parameters on a training corpus by using maximum likelihood estimation under (1) and (2). Thus, if $w_i$ denotes the $i$-th word in the training corpus and $T$ the length of the corpus, we seek the word vectors that maximize

$$\frac{1}{T} \sum_{i=1}^T \sum_{\substack{\delta = -\Delta \\ \delta \neq 0}}^{\Delta} \log p(w_{i+\delta} | w_i) \ . \qquad (3)$$

As mentioned, the normalized context vectors obtained from maximizing (3) under (1) and (2) exhibit additive compositionality. For example, the cosine distance between the sum of the context vectors of the words "Vietnam" and "capital" and the context vector of the word "Hanoi" is small.

While there has been much interest in using algebraic operations on word vectors to carry out semantic operations like composition, the only published work which attempts a rigorous theoretical understanding of this phenomenon is (Arora et al., 2016). This work guarantees that word vectors can be recovered by factorizing the so-called PMI matrix, and that algebraic operations on these word vectors can be used to solve analogies, under certain conditions on the process that generated the training corpus. Specifically, the word vectors must be known, *a priori* before their recovery, to have been generated by randomly scaling uniformly sampled vectors from the unit sphere[2]. Further, the $i$th word in the corpus must have been selected with probability proportional to $e^{\mathbf{u}_w^T \mathbf{c}_i}$, where the "discourse" vector $\mathbf{c}_i$ governs the topic of the corpus at the $i$th word. The discourse vector is assumed to evolve according to a random walk on the unit sphere that has a uniform stationary distribution.

By way of contrast, our results assume nothing about the properties of the word vectors *a priori*. In fact, the connection we establish between the Skip-gram and Sufficient Dimensionality Reduction model of (Globerson and Tishby, 2003)

---

[2]More generally, it suffices that the word vectors have certain properties consistent with this sampling process.

shows that the word vectors learned by Skip-Gram are information-theoretically optimal. Further, the context word $c$ in the Skip-Gram model essentially serves the role that the discourse vector does in the PMI model of (Arora et al., 2016): the words neighboring $c$ are selected with probability proportional to $e^{\mathbf{u}_c^T \mathbf{v}_w}$. We find the exact non-linear composition operator when no assumptions are made on the context word. When an analogous assumption to that of (Arora et al., 2016) is made, that the context words are uniformly distributed, we prove that the composition operator reduces to vector addition.

While our primary motivation has been to provide a better theoretical understanding of the popular SG model, our connection with the SDR method opens up the possibility of practical applicability of our approach more generally. In particular, there is the question of whether, for a given corpus, fitting an SG model will give good embeddings—while we are making reasonable linguistic assumptions about how to model words and the interdependencies of words in a corpus, it's not clear that these have to hold universally on all corpuses to which we apply SG. However, the fact that when we fit an SG model we are fitting an SDR model (up to frequency information), and the fact that SDR models are information-theoretically optimal in a certain sense, argues that regardless of whether the SG assumptions hold, SG always gives us optimal features in the following sense: the learned context embeddings and target embeddings preserve the maximal amount of mutual information between any pair of random variables $X$ and $Y$ consistent with the observed co-occurence matrix, where $Y$ is the target word and $X$ is the predictor word (in a min-max sense, since there are many ways of coupling $X$ and $Y$, each of which may have different amounts of mutual information). Importantly, this statement requires no assumptions on the distribution $P(X, Y)$.

## 2 Compositionality of Skip-Gram

In this section, we first give a mathematical formulation of the intuitive notion of compositionality of words. We then prove that the composition operator for the Skip-Gram model in full generality is a non-linear function of the vectors of the words being composed. Under a single simplifying assumption, the operator *linearizes* and reduces to

the addition of the word vectors. Finally, we explain how linear compositionality allows for solving word analogies with vector algebra.

A natural way of capturing the compositionality of words is to say that the *set* of context words $c_1, \ldots, c_m$ has the same meaning as the single word $c$ if for every other word $w$,

$$p(w|c_1, \ldots, c_m) = p(w|c) \ .$$

Although this is an intuitively satisfying definition, we never expect it to hold exactly; instead, we replace exact equality with the minimization of KL-divergence. That is, we state that the best candidate for having the same meaning as the set of context words $C$ is the word

$$\arg\min_{c \in V} \mathrm{D_{KL}}(p(\cdot|C) \,|\, p(\cdot|c)) \ . \qquad (4)$$

We refer to any vector that minimizes (4) as a *synonym* of the set of words $C$.

There are two natural concerns with (4). The first is that, in general, it is not clear how to define $p(\cdot|C)$. The second is that KL-divergence minimization is a hard problem, as it involves optimization over many high dimensional probability distributions. Our main result shows that both of these problems go away for any language model that satisfies two assumptions:

A1. For every word $c$, there exists $Z_c$ such that for every word $w$,

$$p(w|c) = \frac{1}{Z_c} \exp(\mathbf{u}_c^T \mathbf{v}_w) \ . \qquad (5)$$

A2. For every set of words $C = \{c_1, c_2, \ldots, c_m\}$, there exists $Z_C$ such that for every word $w$,

$$p(w|C) = \frac{p(w)^{1-m}}{Z_C} \prod_{i=1}^{m} p(w|c_i) \ . \qquad (6)$$

Clearly, the SG model satisfies A1 by definition. We prove that it also satisfies A2 when $m \leq \Delta$ (Lemma 1).

**Theorem 1.** *In every word model that satisfies A1 and A2, for every set of words $C = \{c_1, \ldots, c_m\}$, the synonyms of $C$ satisfy*

$$\sum_{w \in V} p(w|c)\mathbf{v}_w = \sum_{w \in V} p(w|C)\mathbf{v}_w \ . \qquad (7)$$

Theorem 1 characterizes the composition operator for any language model which satisfies our two assumptions; in general, this operator is *not* addition. Instead, a synonym $c$ is a vector such that the average word vector under $p(\cdot|c)$ matches that under $p(\cdot|C)$. When the expectations in (7) can be computed, the composition operator can be implemented by solving a non-linear system of equations to find a vector $\mathbf{u}$ for which the left-hand side of (7) equals the right-hand side.

Our next result proves that although the composition operator is nontrivial in the general case, to recover vector addition as the composition operator, it suffices to assume that the word frequency is *uniform*.

**Theorem 2.** *In every word model that satisfies A1, A2, and where $p(w) = 1/|V|$ for every $w \in V$, the synonym of $C = \{c_1, \ldots, c_m\}$ is*

$$\mathbf{u}_1 + \ldots + \mathbf{u}_m \ .$$

As word frequencies are typically much closer to a Zipf distribution (Piantadosi, 2014), the uniformity assumption of Theorem 2 is not realistic. That said, we feel it is important to point out that, as reported in (Mikolov et al., 2013b), additivity captures compositionality *more* accurately when the training set is manipulated so that the prior distribution of the words is made *closer* to uniform.

**Using composition to solve analogies** It has been observed that word vectors trained using nonlinear models like SG tend to encode semantic relationships between words as linear relationships between the word vectors (Mikolov et al., 2013b; Pennington et al., 2014; Levy and Goldberg, 2014). In particular, analogies of the form "man:woman::king:?" can often be solved by taking ? to be the word in the vocabulary whose context vector has the smallest angle with $\mathbf{u}_{\text{woman}} + (\mathbf{u}_{\text{king}} - \mathbf{u}_{\text{man}})$. Theorems 1 and 2 offer insight into the solution such analogy questions.

We first consider solving an analogy of the form "$m$:$w$::$k$:?"" in the case where the composition operator is nonlinear. The fact that $m$ and $w$ share a relationship means $m$ is a synonym for the set of words $\{w, R\}$, where $R$ is a set of words encoding the relationship between $m$ and $w$. Similarly, the fact that $k$ and ? share the same relationship means $k$ is a synonym for the set of words $\{?, R\}$.

By Theorem 1, we have that $R$ and ? must satisfy

$$\sum_{\ell \in V} p(\ell|m)v_\ell = \sum_{\ell \in V} p(\ell|w, R)v_\ell \quad \text{and}$$
$$\sum_{\ell \in V} p(\ell|k)v_\ell = \sum_{\ell \in V} p(\ell|?, R)v_\ell.$$

We see that solving analogies when the composition operator is nonlinear requires the solution of two highly nonlinear systems of equations. In sharp contrast, when the composition operator is linear, the solution of analogies delightfully reduces to elementary vector algebra. To see this, we again begin with the assertion that the fact that $m$ and $w$ share a relationship means $m$ is a synonym for the set of words $\{w, R\}$; Similarly, $k$ is a synonym for $\{?, R\}$. By Theorem 2,

$$\mathbf{u}_m = \mathbf{u}_w + \mathbf{u}_r \quad \text{and}$$
$$\mathbf{u}_k = \mathbf{u}_? + \mathbf{u}_r,$$

which gives the expected relationship

$$\mathbf{u}_? = \mathbf{u}_k + (\mathbf{u}_w - \mathbf{u}_m).$$

Note that because this expression for $\mathbf{u}_?$ is in terms of $k$, $w$, and $m$, there is actually no need to assume that $R$ is a set of actual words in $V$.

## 2.1 Proofs

*Proof of Theorem 1.* Note that $p(w|C)$ equals

$$\frac{p(w)^{1-m}}{Z_C} \prod_{i=1}^{m} p(w|c_i)$$
$$= \frac{p(w)^{1-m}}{Z_C} \exp\left(\sum_{i=1}^{m} \mathbf{u}_{c_i}^T \mathbf{v}_w - \sum_{i=1}^{m} \log Z_{c_i}\right)$$
$$= \frac{1}{Z} p(w)^{1-m} \exp(\mathbf{u}_C^T \mathbf{v}_w) \ ,$$

where $Z = Z_C \prod_{i=1}^{m} Z_i$, and $\mathbf{u}_C = \sum_{i=1}^{m} \mathbf{u}_i$.

Minimizing the KL-divergence

$$D_{\text{KL}}(p(\cdot|c_1, \ldots, c_m)\|p(\cdot|c))$$

as a function of $c$ is equivalent to maximizing the negative cross-entropy as a function of $\mathbf{u}_c$, i.e., as maximizing

$$Q(\mathbf{u}_c) = Z \sum_w \frac{\exp(\mathbf{u}_C^T \mathbf{v}_w)}{p(w)^{m-1}} (\mathbf{u}_c^T \mathbf{v}_w - \log Z_c) \ .$$

Since $Q$ is concave, the maximizers occur where its gradient vanishes. As $\nabla_{\mathbf{u}_c} Q$ equals

$$Z \sum_w \frac{\exp(\mathbf{u}_C^T \mathbf{v}_w)}{p(w)^{m-1}} \left[ \mathbf{v}_w - \frac{\sum_{\ell=1}^n \exp(\mathbf{u}_c^T \mathbf{v}_\ell) \mathbf{v}_\ell}{\sum_{k=1}^n \exp(\mathbf{u}_c^T \mathbf{v}_k)} \right]$$

$$= \frac{\sum_{\ell=1}^n \exp(\mathbf{u}_c^T \mathbf{v}_\ell) \mathbf{v}_\ell}{\sum_{k=1}^n \exp(\mathbf{u}_c^T \mathbf{v}_k)} - Z \sum_w \frac{\exp(\mathbf{u}_C^T \mathbf{v}_w) \mathbf{v}_w}{p(w)^{m-1}}$$

$$= \sum_{w \in V} p(w|c) \mathbf{v}_w - \sum_{w \in V} p(w|c_1, \ldots, c_m) \mathbf{v}_w \ ,$$

we see that (7) follows. $\qquad\square$

*Proof of Theorem 2.* Recall that $\mathbf{u}_C = \sum_{i=1}^m \mathbf{u}_i$. When $p(w) = 1/|V|$ for all $w \in V$, the negative cross-entropy simplifies to

$$Q(\mathbf{u}_c) = Z \sum_w \exp\left(\mathbf{u}_C^T \mathbf{v}_w\right) \left(\mathbf{u}_c^T \mathbf{v}_w - \log Z_c\right) \ ,$$

and its gradient $\nabla_{\mathbf{u}_c} Q$ to

$$Z \sum_w \exp(\mathbf{u}_C^{\ T} \mathbf{v}_w) \left[ \mathbf{v}_w - \frac{\sum_{\ell=1}^n \exp(\mathbf{u}_c^T \mathbf{v}_\ell) \mathbf{v}_\ell}{\sum_{k=1}^n \exp(\mathbf{u}_c^T \mathbf{v}_k)} \right]$$

$$= Z \sum_w \exp(\mathbf{u}_C^{\ T} \mathbf{v}_w) \mathbf{v}_w - \sum_w \exp(\mathbf{u}_c^T \mathbf{v}_w) \mathbf{v}_w \ .$$

Thus, $\nabla Q(\mathbf{u}_C) = 0$ and since $Q$ is concave, $\mathbf{u}_C$ is its unique maximizer. $\qquad\square$

**Lemma 1.** *The SG model satisfies assumption A2 when $m \le \Delta$.*

*Proof of Lemma 1.* First, assume that $m = \Delta$. In the SG model target words are conditionally independent given a context word, i.e.,

$$p(c_1, \ldots, c_m | w) = \prod_{i=1}^m p(c_i | w).$$

Applying Baye's rule,

$$p(w|c_1, \ldots, c_m) = \frac{p(c_1, \ldots, c_m | w) p(w)}{p(c_1, \ldots, c_m)}$$

$$= \frac{p(w)}{p(c_1, \ldots, c_m)} \prod_{i=1}^m p(c_i | w)$$

$$= \frac{p(w)}{p(c_1, \ldots, c_m)} \prod_{i=1}^m \frac{p(w|c_i) p(c_i)}{p(w)}$$

$$= \frac{p(w)^{1-m}}{Z_C} \prod_{i=1}^m p(w|c_i) \ , \qquad (8)$$

where $Z_C = 1/\left(\prod_{i=1}^m p(c_i)\right)$. This establishes the result when $m = \Delta$. The cases $m < \Delta$ follow by marginalizing out $\Delta - m$ context words in the equality 8. $\qquad\square$

**Projection of synonyms onto the vocabulary** Theorem 2 states that if there is a word $c$ in the vocabulary $V$ whose context vector equals the sum of the context vectors of the words $c_1, \ldots, c_m$, then $c$ has the same "meaning", in the sense of (4), as the composition of the words $c_1, \ldots, c_m$. For any given set of words $C = \{c_1, \ldots, c_m\}$, it is unlikely that there exists a word $c \in V$ whose context vector is exactly equal to the sum of the context vectors of the words $c_1, \ldots, c_m$. Similarly, in Theorem 1, the solution(s) to (7) will most likely not equal the context vector of any word in $V$. In both cases, we thus need to project the vector(s) onto words in our vocabulary in some manner.

Since Theorem 1 holds for any prior over $V$, in theory, we could enumerate all words in $V$ and find the word(s) that minimize the difference of the left hand side of (7) from the right hand side. In practice, it turns out that the *angle* between the context vector of a word $w \in V$ and solution-vector(s) is a good proxy and one gets very good experimental results by selecting as the synonym of a collection of words, the word that minimizes the angle to the synonym vector.

Minimizing the angle has been empirically successful at capturing composition in multiple log-linear word models. One way to understand the success of this approach is to recall that each word $c$ is characterized by a categorical distribution over all other words $w$, as stated in (1). The peaks of this categorical distribution are precisely the words with which $c$ co-occurs most often. These words characterize $c$ more than all the other words in the vocabulary, so it is reasonable to expect that a word $c'$ whose categorical distribution has similar peaks as the categorical distribution of $c$ is similar in meaning to $c$. Note that the location of the peaks of $p(\cdot|c)$ are immune to the scaling of $\mathbf{u}_c$ (although the values of $p(\cdot|c)$ may change); thus, the words $w$ which best characterize $c$ are those for which $\mathbf{v}_w$ has a high inner product with $\mathbf{u}_c/\|\mathbf{u}_c\|_2$. Since

$$\left| \frac{\mathbf{u}_c^T \mathbf{v}_w}{\|\mathbf{u}_c\|_2} - \frac{\mathbf{u}_{c'}^T \mathbf{v}_w}{\|\mathbf{u}_{c'}\|_2} \right| \le \sqrt{2 \left( 1 - \frac{\mathbf{u}_c^T \mathbf{u}_{c'}}{\|\mathbf{u}_c\|_2 \|\mathbf{u}_{c'}\|_2} \right)} \|\mathbf{v}_w\|_2,$$

it is clear that if the angle between the context representations of $c$ and $c'$ is small, the distributions $p(w|c)$ and $p(w|c')$ will tend to have similar peaks.

## 3 Skip-Gram learns a Sufficient Dimensionality Reduction Model

The Skip-Gram model assumes that the distribution of the neighbors of a word follows a specific exponential parametrization of a categorical distribution. There is empirical evidence that this model generates features that are useful for NLP tasks, but there is no *a priori* guarantee that the training corpus was generated in this manner. In this section, we provide theoretical support for the usefulness of the features learned even when the Skip-Gram model is misspecified.

To do so, we draw a connection between Skip-Gram and the Sufficient Dimensionality Reduction (SDR) factorization of Globerson and Tishby (Globerson and Tishby, 2003). The SDR model learns optimal[3] embeddings for discrete random variables $X$ and $Y$ *without assuming any parametric form on the distributions* of $X$ and $Y$, and it is useful in a variety of applications, including information retrieval, document classification, and association analysis (Globerson and Tishby, 2003). As it turns out, these embeddings, like Skip-Gram, are obtained by learning the parameters of an exponentially parameterized distribution. In Theorem 3 below, we show that if a Skip-Gram model is fit to the cooccurence statistics of $X$ and $Y$, then the output can be trivially modified (by adding readily-available information on word frequencies) to obtain the parameters of an SDR model.

This connection is significant for two reasons: first, the original algorithm of (Globerson and Tishby, 2003) for learning SDR embeddings is expensive, as it involves information projections. Theorem 3 shows that if one can efficiently fit a Skip-Gram model, then one can efficiently fit an SDR model. This implies that Skip-Gram specific approximation heuristics like negative-sampling, hierarchical softmas, and Glove, which are believed to return high-quality approximations to Skip-Gram parameters (Mikolov et al., 2013b; Pennington et al., 2014), can be used to efficiently approximate SDR model parameters. Second, (Globerson and Tishby, 2003) argues for the optimality of the SDR embedding in any domain where the training information on $X$ and $Y$

---

[3]Optimal in an information-theoretic sense: they preserve the maximal mutual information between any pair of random variables with the observed coocurence statistics, without regard to the underlying joint distribution.

consists of their coocurrence statistics; this optimality and the Skip-Gram/SDR connection argues for the use of Skip-Gram approximations in such domains, and supports the positive experimental results that have been observed in applications in network science (Grover and Leskovec, 2016), proteinomics (Asgari and Mofrad, 2015), and other fields.

As stated above, the SDR factorization solves the problem of finding information-theoretically optimal features, given co-occurrence statistics for a pair of discrete random variables $X$ and $Y$. Associate a vector $\mathbf{w}_i$ to the $i$th state of $X$, a vector $\mathbf{h}_j$ to the $j$th state of $Y$, and let $\mathbf{W} = [\mathbf{w}_1^T \cdots \mathbf{w}_{|X|}^T]^T$ and $\mathbf{H}$ be defined similarly. Globerson and Tishby show that such optimal features can be obtained from a low-rank factorization of the matrix $\mathbf{G}$ of co-occurence measurements: $G_{ij}$ counts the number of times state $i$ of $X$ has been observed to co-occur with state $j$ of $Y$. The loss of this factorization is measured using the KL-divergence, and so the optimal features are obtained from solving the problem

$$\arg\min_{\mathbf{W},\mathbf{H}} \mathrm{D_{KL}} \left( \frac{\mathbf{G}}{Z_{\mathbf{G}}} \,\middle\|\, \frac{1}{Z_{\mathbf{W},\mathbf{H}}} e^{\mathbf{W}\mathbf{H}^T} \right).$$

Here, $Z_{\mathbf{G}} = \sum_{ij} G_{ij}$ normalizes $\mathbf{G}$ into an estimate of the joint pmf of $X$ and $Y$, and similarly $Z_{\mathbf{W},\mathbf{H}}$ is the constant that normalizes $e^{\mathbf{W}\mathbf{H}^T}$ into a joint pmf. The expression $e^{\mathbf{W}\mathbf{H}^T}$ denotes entry-wise exponentiation of $\mathbf{W}\mathbf{H}^T$.

Now we revisit the Skip-Gram training objective, and show that it differs from the SDR objective only slightly. Whereas the SDR objective measures the distance between the pmfs given by (normalized versions of) $\mathbf{G}$ and $e^{\mathbf{W}\mathbf{H}^T}$, the Skip-Gram objective measures the distance between the pmfs given by (normalized versions of) *the rows* of $\mathbf{G}$ and $e^{\mathbf{W}\mathbf{H}^T}$. That is, SDR emphasizes fitting the entire pmfs, while Skip-Gram emphasizes fitting conditional distributions.

Before presenting our main result, we state and prove the following lemma, which is of independent interest and is used in the proof of our main theorem. Recall that Skip-Gram represents each word $c$ as a multinomial distribution over all other words $w$, and it learns the parameters for these distributions by a maximum likelihood estimation. It is known that learning model parameters by maximum likelihood estimation is equivalent to minimizing the KL-divergence of the learned

model from the empirical distribution; the following lemma establishes the KL-divergence that Skip-Gram minimizes.

**Lemma 2.** *Let* $\mathbf{G}$ *be the word co-occurrence matrix constructed from the corpus on which a Skip-Gram model is trained, in which case* $G_{cw}$ *is the number of times word* $w$ *occurs as a neighboring word of* $c$ *in the corpus. For each word* $c$*, let* $g_c$ *denote the empirical frequency of the word in the corpus, so that*

$$g_c = \sum_w G_{cw} / \sum_{t,w} G_{t,w}.$$

*Given a positive vector* $\mathbf{x}$*, let* $\hat{\mathbf{x}} = \mathbf{x}/\|\mathbf{x}\|_1$*. Then, the Skip-Gram model model parameters* $\mathbf{U} = \begin{bmatrix} \mathbf{u}_1 & \cdots & \mathbf{u}_{|V|} \end{bmatrix}^T$ *and* $\mathbf{V} = \begin{bmatrix} \mathbf{v}_1 & \cdots & \mathbf{u}_{|V|} \end{bmatrix}^T$ *minimize the objective*

$$\sum_c g_c \mathrm{D_{KL}}(\hat{\mathbf{g}}^c \,\|\, \widehat{e^{\mathbf{u}_c^T \mathbf{V}^T}}),$$

*where* $\mathbf{g}^c$ *is the cth row of* $G$*.*

*Proof.* Recall that Skip-Gram chooses $\mathbf{U}$ and $\mathbf{V}$ to maximize

$$Q = \frac{1}{T} \sum_{i=1}^{T} \sum_{\substack{\delta=-C \\ \delta \neq 0}}^{C} \log p(w_{i+\delta}|w_i) \ ,$$

where

$$p(w|c) = \frac{e^{\mathbf{u}_c^T \mathbf{v}_w}}{\sum_{i=1}^{n} e^{\mathbf{u}_c^T \mathbf{v}_i}} \ .$$

This objective can be rewritten using the pairwise cooccurence statistics as

$$Q = \frac{1}{T} \sum_{c,w} G_{cw} \log p(w|c)$$

$$= \frac{1}{T} \sum_c \left[ \left( \sum_t G_{ct} \right) \sum_w \frac{G_{cw}}{\sum_t G_{ct}} \log p(w|c) \right]$$

$$\propto \frac{1}{T} \sum_c \left[ \frac{(\sum_t G_{ct})}{(\sum_{tw} G_{tw})} \sum_w \frac{G_{cw}}{\sum_t G_{ct}} \log p(w|c) \right]$$

$$= \sum_c g_c \left( \sum_w (\hat{\mathbf{g}}^c)_w \log p(w|c) \right)$$

$$= \sum_c g_c \left( -\mathrm{D_{KL}}(\hat{\mathbf{g}}^c \,\|\, p(\cdot|c)) - H(\hat{\mathbf{g}}^c) \right),$$

where $H(\cdot)$ denotes the entropy of a distribution. It follows that since Skip-Gram maximizes $Q$, it minimizes

$$\sum_c g_c \mathrm{D_{KL}}(\hat{\mathbf{g}}^c \,\|\, p(\cdot|c)) = \sum_c g_c \mathrm{D_{KL}}(\hat{\mathbf{g}}^c \,\|\, \widehat{e^{\mathbf{u}_c^T \mathbf{V}^T}}).$$

$\square$

We now prove our main theorem of this section, which states that SDR parameters can be obtained by augmenting the Skip-Gram embeddings to account for word frequencies.

**Theorem 3.** *Let* $\mathbf{U}, \mathbf{V}$ *be the results of fitting a Skip-Gram model to* $\mathbf{G}$*, and consider the augmented matrices*

$$\tilde{\mathbf{U}} = [\mathbf{U} \,|\, \boldsymbol{\alpha}] \text{ and } \tilde{\mathbf{V}} = [\mathbf{V} \,|\, \mathbf{1}],$$

*where*

$$\alpha_c = \log \left( \frac{g_c}{\sum_w e^{\mathbf{u}_c^T \mathbf{v}_w}} \right) \text{ and } g_c = \frac{\sum_w G_{c,w}}{\sum_{t,w} G_{t,w}}.$$

*Then, the features* $(\tilde{\mathbf{U}}, \tilde{\mathbf{V}})$ *constitute a sufficient dimensionality reduction of* $\mathbf{G}$*.*

*Proof.* For convenience, let $\overline{\mathbf{G}}$ denote the joint pdf matrix $\mathbf{G}/Z_{\mathbf{G}}$, and let $\widehat{\mathbf{G}}$ denote the matrix obtained by normalizing each row of $\mathbf{G}$ to be a probability distribution. Then, it suffices to show that $D_{KL}(\overline{\mathbf{G}} \,\|\, q_{\mathbf{W},\mathbf{H}})$ is minimized over the set of probability distributions

$$\left\{ q_{\mathbf{W},\mathbf{H}} \,\middle|\, q_{\mathbf{W},\mathbf{H}}(w,c) = \frac{1}{Z} \left( e^{\mathbf{W}\mathbf{H}^T} \right)_{cw} \right\},$$

when $\mathbf{W} = \tilde{\mathbf{U}}$ and $\mathbf{H} = \tilde{\mathbf{V}}$.

To establish this result, we use a chain rule for the KL-divergence. Recall that if we denote the expected KL-divergence between two marginal pmfs by

$$D_{KL}(p(\cdot|c)\|q(\cdot|c))$$

$$= \sum_c p(c) \left( \sum_w p(w|c) \log \left( \frac{p(w|c)}{q(w|c)} \right) \right),$$

then the KL-divergence satisfies the chain rule:

$$D_{KL}(p(w,c)\|q(w,c))$$
$$= D_{KL}(p(c)\|q(c)) + D_{KL}(p(w|c)\|q(w|c)).$$

Using this chain rule, we get

$$D_{KL}(\overline{\mathbf{G}} \,\|\, q_{\mathbf{W},\mathbf{H}}(w,c)) \qquad (9)$$
$$= D_{KL}(\mathbf{g} \,\|\, q_{\mathbf{W},\mathbf{H}}(c)) + D_{KL}(\widehat{\mathbf{G}} \| q_{\mathbf{W},\mathbf{H}}(w|c)).$$

Note that the second term in this sum is, in the notation of Lemma 2,

$$D_{KL}(\widehat{\mathbf{G}} \| q_{\mathbf{W},\mathbf{H}}(w|c)) = \sum_c g_c \mathrm{D_{KL}}(\hat{\mathbf{g}}^c \,\|\, \widehat{e^{\mathbf{w}_c^T \mathbf{H}^T}}),$$

so the matrices $\mathbf{U}$ and $\mathbf{V}$ that are returned by fitting the Skip-Gram model minimize the second term in this sum. We now show that the augmented matrices $\mathbf{W} = \tilde{\mathbf{U}}$ and $\mathbf{H} = \tilde{\mathbf{V}}$ also minimize this second term, and in addition they make the first term vanish.

To see that the first of these claims holds, i.e., that the augmented matrices make the second term in (9) vanish, note that

$$q_{\tilde{\mathbf{U}},\tilde{\mathbf{V}}}(w|c) \propto e^{\tilde{\mathbf{u}}_c^T \tilde{\mathbf{v}}_w} = e^{\mathbf{u}_c^T \mathbf{v}_w + \alpha_c} \propto q_{\mathbf{U},\mathbf{V}}(w|c),$$

and the constant of proportionality is independent of $w$. It follows that $q_{\tilde{\mathbf{U}},\tilde{\mathbf{V}}}(w|c) = q_{\mathbf{U},\mathbf{V}}(w|c)$ and

$$D_{KL}(\widehat{\mathbf{G}} \,\|\, q_{\tilde{\mathbf{U}},\tilde{\mathbf{V}}}(w|c)) = D_{KL}(\widehat{\mathbf{G}} \,\|\, q_{\mathbf{U},\mathbf{V}}(w|c)).$$

Thus, the choice $\mathbf{W} = \tilde{\mathbf{U}}$ and $\mathbf{H} = \tilde{\mathbf{V}}$ minimizes the second term in (9).

To see that the augmented matrices make the first term in (9) vanish, observe that when $\mathbf{W} = \tilde{\mathbf{U}}$ and $\mathbf{H} = \tilde{\mathbf{V}}$, we have that $q_{\tilde{\mathbf{U}},\tilde{\mathbf{V}}}(c) = \mathbf{g}$ by construction. This can be verified by calculation:

$$q_{\tilde{\mathbf{U}},\tilde{\mathbf{V}}}(c) = \frac{\sum_w q_{\tilde{\mathbf{U}},\tilde{\mathbf{V}}}(w,c)}{\sum_{w,t} q_{\tilde{\mathbf{U}},\tilde{\mathbf{V}}}(w,t)} = \frac{\sum_w e^{\mathbf{u}_c^T \mathbf{v}_w + \alpha_c}}{\sum_{w,t} e^{\mathbf{u}_t^T \mathbf{v}_w + \alpha_t}}$$

$$= \frac{\left(\sum_w e^{\mathbf{u}_c^T \mathbf{v}_w}\right) e^{\alpha_c}}{\sum_t \left(\sum_w e^{\mathbf{u}_t^T \mathbf{v}_w}\right) e^{\alpha_t}}$$

$$= \frac{\left[(e^{\mathbf{U}\mathbf{V}^T} \mathbf{1}) \odot e^{\boldsymbol{\alpha}}\right]_c}{\mathbf{1}^T \left[(e^{\mathbf{U}\mathbf{V}^T} \mathbf{1}) \odot e^{\boldsymbol{\alpha}}\right]}.$$

Here, the notation $\mathbf{x} \odot \mathbf{y}$ denotes entry-wise multiplication of vectors.

Since $\alpha_c = \log(g_c) - \log\left(\left(e^{\mathbf{U}\mathbf{V}^T} \mathbf{1}\right)_c\right)$, we have

$$q_{\tilde{\mathbf{U}},\tilde{\mathbf{V}}}(c) = \frac{\left[(e^{\mathbf{U}\mathbf{V}^T} \mathbf{1}) \odot e^{\boldsymbol{\alpha}}\right]_c}{\mathbf{1}^T \left[(e^{\mathbf{U}\mathbf{V}^T} \mathbf{1}) \odot e^{\boldsymbol{\alpha}}\right]} = \frac{g_c}{\sum_t g_t} = g_c.$$

The choice $\mathbf{W} = \tilde{\mathbf{U}}$ and $\mathbf{H} = \tilde{\mathbf{V}}$ makes the first term in (9) vanish, and it also minimizes the second term in (9). Thus, it follows that the features $(\tilde{\mathbf{U}}, \tilde{\mathbf{V}})$ constitute a sufficient dimensionality reduction of $\mathbf{G}$. $\qquad\square$

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
