# Peer review of "Skip-Gram - Zipf + Uniform = Vector Additivity"

_ACL 2017 — decision unknown_

[Official Review · Reviewer 1 · rating 4 · confidence 4]
soundness 4 · originality 4 · clarity 3 · impact 3 · substance 4 · appropriateness 5 · meaningful comparison 3 · presentation format Oral Presentation

This paper delves into the mathematical properties of the skip-gram model,
explaining the reason for its success on the analogy task and for the general
superiority of additive composition models. It also establishes a link between
skip-gram and Sufficient Dimensionality Reduction.

I liked the focus of this paper on explaining the properties of skip-gram, and
generally found it inspiring to read. I very much appreciate the effort to
understand the assumptions of the model, and the way it affects (or is affected
by) the composition operations that it is used to perform. In that respect, I
think it is a very worthwhile read for the community.

My main criticism is however that the paper is linguistically rather naive. The
authors' use of 'compositionality' (as an operation that takes a set of words
and returns another with the same meaning) is extremely strange. Two words can
of course be composed and produce a vector that is a) far away from both; b)
does not correspond to any other concept in the space; c) still has meaning
(productivity wouldn't exist otherwise!) Compositionality in linguistic terms
simply refers to the process of combining linguistic constituents to produce
higher-level constructs. It does not assume any further constraint, apart from
some vague (and debatable) notion of semantic transparency. The paper's
implication (l254) that composition takes place over sets is also wrong:
ordering matters hugely (e.g. 'sugar cane' is not 'cane sugar'). This is a
well-known shortcoming of additive composition. 

Another important aspect is that there are pragmatic factors that make humans
prefer certain phrases to single words in particular contexts (and the
opposite), naturally changing the underlying distribution of words in a large
corpus. For instance, talking of a 'male royalty' rather than a 'king' or
'prince' usually has implications with regard to the intent of the speaker
(here, perhaps highlighting a gender difference). This means that the equation
in l258 (or for that matter the KL-divergence modification) does not hold, not
because of noise in the data, but because of fundamental linguistic processes.
This point may be addressed by the section on SDR, but I am not completely sure
(see my comments below).

In a nutshell, I think the way that the authors present composition is flawed,
but the paper convinces me that this is indeed what happens in skip-gram, and I
think this is an interesting contribution. 

The part about Sufficient Dimensionality Reduction seems a little disconnected
from the previous argument as it stands. I'm afraid I wasn't able to fully
follow the argument, and I would be grateful for some clarification in the
authors' response. If I understand it well, the argument is that skip-gram
produces a model where a word's neighbours follow some exponential
parametrisation of a categorical distribution, but it is unclear whether this
actually reflects the distribution of the corpus (as opposed to what happens
in, say, a pure count-based model). The fact that skip-gram performs well
despite not reflecting the data is that it implements some form of SDR, which
does not need to make any assumption about the underlying form of the data. But
then, is it fair to say that the resulting representations are optimised for
tasks where geometrical regularities are important, regardless of the actual
pattern of the data? I.e. there some kind of denoising going on?

Minor comments:

- The abstract is unusually long and could, I think, be shortened.

- para starting l71: I think it would be misconstrued to see circularity here.
Firth observed that co-occurrence effects were correlated with similarity
judgements, but those judgements are the very cognitive processes that we are
trying to model with statistical methods. Co-occurrence effects and vector
space word representations are in some sense 'the same thing', modelling an
underlying linguistic process we do not have direct observations for. So
pair-wise similarity is not there to break any circularity, it is there because
it better models the kind of judgements humans known to make.

- l296: I think 'paraphrase' would be a better word than 'synonym' here, given
that we are comparing a set of words with a unique lexical item.

- para starting l322: this is interesting, and actually, a lot of the zipfian
distribution (the long tail) is fairly uniform.

- l336: it is probably worth pointing out that the analogy relation does not
hold so well in practice and requires to 'ignore' the first returned neighbour
of the analogy computation (which is usually one of the observed terms).

- para starting l343: I don't find it so intuitive to say that 'man' would be a
synonym/paraphrase of anything involving 'woman'. The subtraction involved in
the analogy computation is precisely not a straightforward composition
operation, as it involves an implicit negation. 

- A last, tiny general comment. It is usual to write p(w|c) to mean the
probability of a word given a context, but in the paper 'w' is actually the
context and 'c' the target word. It makes reading a little bit harder...
Perhaps change the notation?

Literature:

The claim that Arora (2016) is the only work to try and understand vector
composition is a bit strong. For instance, see the work by Paperno & Baroni on
explaining the success of addition as a composition method over PMI-weighted
vectors:

D. Paperno and M. Baroni. 2016. When the whole is less than the sum of its
parts: How composition affects PMI values in distributional semantic vectors.
Computational Linguistics 42(2): 345-350.

***
I thank the authors for their response and hope to see this paper accepted.